# Astrocytic CD44 Deficiency Reduces the Severity of Kainate-Induced Epilepsy

**DOI:** 10.3390/cells12111483

**Published:** 2023-05-26

**Authors:** Patrycja K. Kruk, Karolina Nader, Anna Skupien-Jaroszek, Tomasz Wójtowicz, Anna Buszka, Gabriela Olech-Kochańczyk, Grzegorz M. Wilczynski, Remigiusz Worch, Katarzyna Kalita, Jakub Włodarczyk, Joanna Dzwonek

**Affiliations:** 1Laboratory of Cell Biophysics, Nencki Institute of Experimental Biology, Polish Academy of Sciences, 3 Pasteura St, 02-093 Warsaw, Poland; 2Laboratory of Neurobiology, Nencki-EMBL Partnership for Neural Plasticity and Brain Disorders-Braincity, 3 Pasteura St, 02-093 Warsaw, Poland; 3Laboratory of Molecular and Structural Neuromorphology, Nencki Institute of Experimental Biology, Polish Academy of Sciences, 3 Pasteura St, 02-093 Warsaw, Poland

**Keywords:** epilepsy, CD44, astrocytes, synapse, seizures

## Abstract

Background: Epilepsy affects millions of people worldwide, yet we still lack a successful treatment for all epileptic patients. Most of the available drugs modulate neuronal activity. Astrocytes, the most abundant cells in the brain, may constitute alternative drug targets. A robust expansion of astrocytic cell bodies and processes occurs after seizures. Highly expressed in astrocytes, CD44 adhesion protein is upregulated during injury and is suggested to be one of the most important proteins associated with epilepsy. It connects the astrocytic cytoskeleton to hyaluronan in the extracellular matrix, influencing both structural and functional aspects of brain plasticity. Methods: Herein, we used transgenic mice with an astrocyte CD44 knockout to evaluate the impact of the hippocampal CD44 absence on the development of epileptogenesis and ultrastructural changes at the tripartite synapse. Results: We demonstrated that local, virally-induced CD44 deficiency in hippocampal astrocytes reduces reactive astrogliosis and decreases the progression of kainic acid-induced epileptogenesis. We also observed that CD44 deficiency resulted in structural changes evident in a higher dendritic spine number along with a lower percentage of astrocyte-synapse contacts, and decreased post-synaptic density size in the hippocampal molecular layer of the dentate gyrus. Conclusions: Overall, our study indicates that CD44 signaling may be important for astrocytic coverage of synapses in the hippocampus and that alterations of astrocytes translate to functional changes in the pathology of epilepsy.

## 1. Introduction

Epilepsy is one of the most common chronic neurological disorders, affecting around 50 million people worldwide. It is characterized by recurrent seizures which are a result of excessive electrical discharges in neurons [1]. Most of the currently available antiepileptic drugs work by suppressing neuronal hyperexcitability, but they also cause several side effects, negatively affecting patients’ quality of life [2]. Additionally, one-third of patients suffer from drug-resistant types of epilepsy [3], of which temporal lobe epilepsy (TLE) is the most common one. A target alternative to neurons, such as astrocytes able to modulate neuronal activity, could be a promising next step in the treatment of epilepsy. Reactive astrocytes in the epileptic brain are well known to undergo extensive morphological and biochemical changes that modify their physiological functions, including glutamate reuptake, regulation of the ionic environment and interstitial volume, and blood-brain barrier permeability control [4,5]. Both the protective and detrimental functions of reactive astrocytes in seizure development have been documented [6,7]. A better understanding of the pro- and anticonvulsant actions of reactive astrocytes could lead to novel strategies for epilepsy treatment.

Seizure onset in TLE can begin in the interconnected structures within the temporal lobe and adjoining regions, such as the hippocampus [8]. Epileptic seizures induce morphological rearrangements of neuronal circuits that are accompanied by pruning, remodeling, and de novo formation of synapses in the hippocampus [9,10]. The loss of neurons in the hippocampus has been described in both human specimens [11] and animal models [12,13,14,15] of TLE. Experimental animal models of *status epilepticus* (SE) showed that hippocampal dendritic spines are dynamically altered during the establishment and maintenance of spontaneous seizures. In addition to the extensive loss of spines, the remaining spines are of increased size and often appear as multisynaptic giant spines, with numerous postsynaptic densities (PSDs), making contact with different presynaptic boutons [9,16,17,18].

Astrocytes possess thin terminal processes, also called perisynaptic astrocyte processes (PAPs) or leaflets, that surround synapses and regulate synaptic transmission by bilateral communication (reviewed in [19,20]). PAPs located close to a synaptic cleft undergo structural rearrangements upon neuronal activation, which results in altering synaptic transmission and plasticity [21,22,23,24]. Interestingly, astrocytes also undergo seizure-induced alterations characterized by overall hypertrophy [25,26], and their ensheathment around presynaptic and postsynaptic elements is enhanced in CA3-CA1 synapses after kainate-induced SE [27]. However, little is known about the functional relevance of morphological changes in astrocytes following SE.

CD44 protein, the main transmembrane receptor for hyaluronan (HA), is expressed in the central nervous system (CNS). The localization of CD44 protein in the brain varies depending on cell type and brain region and is developmentally regulated. A number of reports have pointed to the astrocytic expression of CD44 in the nervous system [28,29,30,31], while others have also described its expression in neurons [32,33,34,35,36]. The diversity of the CD44 cell type and developmental expression should be considered when interpreting data concerning its function. While the extracellular N-terminal domain of CD44 is responsible for the binding of HA, the C-terminal cytoplasmatic region is involved in interactions with different effectors, along with cytoskeletal proteins [37]. CD44-mediated signaling can additionally influence the function of small Rho GTPases (Rac1-Rac Family Small GTPase 1, Cdc42-Cell Division Cycle 42, RhoA-Ras Homolog Family Member A) [32,38,39] and Src kinases [33]. CD44 is upregulated during injury and other pathological conditions, including epilepsy [40,41,42,43]. Its strong reactivity is seen in both pilocarpine- and kainate-induced SE, as well as various epilepsy-related neurological disorders [42,44]. On the other hand, a huge increase in the number of CD44-positive astrocytes was observed in the post-mortem brains of Alzheimer’s disease patients, which was not related to epilepsy [43]. In humans, transcriptome analysis in epileptic patients identified CD44 as one of the five most important proteins associated with the pathogenesis of epilepsy [45].

CD44 expression correlates with the morphological characteristics of astrocytes. Our previous in vitro studies demonstrated that astrocytes without CD44 adhesion protein have distinctive stellate-like morphology [39]. Similarly, CD44-negative glial cells in the human brain show numerous branches and short processes and are often classified as protoplasmic astrocytes. In comparison, a high CD44 expression level is observed in more elongated cells with fewer long and unbranched processes; these cells have been categorized as fibrous astrocytes [39,46]. Nevertheless, the role of CD44 in astrocytes in physiological and pathological conditions in the brain, especially during epileptogenesis, remains largely unknown.

In this study, we employed molecular biology techniques, simultaneous EEG and video recording (vEEG), fluorescence immunolabelling, serial block-face scanning electron microscopy (SBEM), and 3D reconstruction [47] to evaluate the impact of the CD44 absence in hippocampal astrocytes on the development of epileptogenesis and consecutive ultrastructural alterations in the tripartite synapse. The presented results point out significant changes in the ultrastructural and functional remodeling of astrocytes and PSD, together with altered seizure progression during epileptogenesis. We show that CD44 deficiency in astrocytes decreases the number of behavioral seizures, increases the number of grade 0 seizures, and reduces reactive astrogliosis during epileptogenesis triggered by kainic acid (KA) administration. Our findings demonstrate that CD44 astrocytic knockout mice show a decreased percentage of astrocyte–synapse contacts, and increased dendritic spine number, along with a smaller PSD area and volume in the molecular layer (ML) of the hippocampal dentate gyrus (DG) of kainic acid-treated mice. Overall, our work suggests that CD44 signaling is involved in astrocyte coverage of neuronal synapses in the ML of the hippocampus and that astrocyte morphological modifications may translate to functional changes in the pathology of epilepsy.

## 2. Materials and Methods

### 2.1. Animals

Adult mice (female and male, 12 to 14 weeks old at the start of the experiments) carrying the conditional deletion of exon 3 of the CD44 gene (CD44*^fl/fl^*; “floxed” mice) were initially acquired from Professor Veronique Orian-Rousseau (Karlsruhe Institute of Technology, Germany) [48]. Animals were bred in the Animal House of the Nencki Institute of Experimental Biology and were treated according to the ethical standards of European and Polish regulations. All mice had free access to food and water, and were kept in individually ventilated cages (at temperatures 21–23 °C, 50–60% humidity) under a 12-h light/12-h dark cycle. All experimental procedures were performed following the European Communities Council Directive of 24 November 1986 (86/609/EEC), Animal Protection Act of Poland, and approved by the 1st Local Ethics Committee in Warsaw (permission no. 860/2019).

### 2.2. Generation of CD44 Knockout Mice

Before the surgery, mice were anesthetized with a mix of ketamine + medetomidine (80 mg/kg + 1 mg/kg, i.p.) and were given anti-inflammatory and analgesic drugs (butorphanol 3.3 mg/kg, tolfenamic acid 2.0 mg/kg). Then, they were placed in a stereotaxic frame on a heating pad to maintain a constant body temperature. To produce animals with the astrocytic knockout of CD44 (AsKO), one group was stereotaxically injected with 500 nL of AAV5 vector with Cre recombinase–AAV5.*gfaABC1D*::GFP-Cre (construct: Bryan Roth, AAV Serotype 5, UNC Vector Core, Chapel Hill, NC, USA, AAV vector with ITR2 where the human *gfaABC1D* promoter [49] drives the expression of eGFP attached to the N-terminus of Cre recombinase), whereas the control (CTRL) group was injected with 500 nL of AAV5.*gfaABC1D*::GFP vector (Construct: Bryan Roth, Addgene plasmid #50473; AAV Serotype 5, UNC Vector Core, AAV vector with ITR2 where the human *gfaABC1D* promoter drives the expression of eGFP). Injections were performed at coordinates (AP −2.0 and ML ±1.4 from bregma, DV −1.5 below the dura) corresponding to the molecular layer of the DG of both hippocampi. The proper injection side was confirmed post-mortem before histochemical analysis. After-surgery recovery time lasted 4 weeks.

### 2.3. Western Blot

The dentate gyrus part of the hippocampus was dissected and homogenized in RIPA lysis buffer with protease inhibitor and phosphatase inhibitor cocktails (Sigma-Aldrich, St. Louis, MO, USA) using TissueLyser II (Qiagen, Hilden, Germany). Protein concentration was measured with a BCA Protein Assay Kit (Pierce, Appleton, WI, USA). Proteins were separated and transferred using 10% TGX FastCast Acrylamide Gels and Trans-Blot Turbo Mini PVDF Transfer Packs (Bio-Rad, Hercules, CA, USA). Western blot was performed according to a standard procedure, with anti-CD44 antibody (R&D Systems, Minneapolis, MN, USA, AF6127; 1:2500) and anti-GAPDH antibody (Abcam, Cambridge, UK, ab9485; 1:5000), with the latter being used as a loading control. For band visualization, the chemiluminescence detection method was used. For densitometric calculations, a scan of the X-ray film was analyzed using the Band/Peak Quantification plugin [50] in Fiji 2.3.0 software [29] and normalized to GAPDH.

### 2.4. Induction of Status Epilepticus

To examine the changes in the CD44 expression level in kainate-induced SE, CD44*^fl/fl^* mice underwent intrahippocampal administration of KA. This model reproduces many of the morphological and functional features similar to human patients with temporal lobe epilepsy [51]. Before the surgery, mice were anesthetized with a mix of ketamine + medetomidine (80 mg/kg + 1 mg/kg, i.p.) and were given anti-inflammatory and analgesic drugs (butorphanol 3.3 mg/kg and tolfenamic acid 2.0 mg/kg). Then, animals were placed in a stereotaxic frame on a heating pad to maintain a constant body temperature. Animals were stereotaxically injected with 70 nL of 20 mM solution of KA (Tocris Bioscience, Bristol, UK) in 0.9% *w*/*v* NaCl saline or saline alone (SA) to the left CA1 region of the dorsal hippocampus (coordinates: AP −1.8, ML +1.7, DV −2.1, relative to bregma). Injections were administered at a 50 nL/min flow rate. Mice were sacrificed 4 days post-surgery.

To induce SE for electroencephalographic monitoring, 4 weeks after induction of the conditional mutation, mice underwent a second surgery. Animals from both AsKO and CTRL groups were stereotaxically injected with KA (AsKO+KA, CTRL+KA) or SA (AsKO, CTRL) as described above. After the surgery, they were given atipamezole hydrochloride (0.5 mg/kg, i.p.) to facilitate awakening from anesthesia.

### 2.5. Electrode Implantation and EEG-Video Monitoring

The implantation of five electrodes was carried out immediately after kainate administration during the same surgery session. Two stainless-steel screw electrodes (1.6 mm Ø; Bilaney Consultants GmbH, Dusseldorf, Germany) were bilaterally positioned over the frontal cortex, with two additional ones over the cerebellum (as the right cerebellar electrode served as the ground electrode, there was a total of three reference electrodes). A bipolar hippocampal electrode (Bilaney Consultants GmbH) was implanted into the left hippocampus (coordinates: AP −2.0 and ML +1.3 from bregma, DV −1.7 below the dura). The placement of the electrode was confirmed post-mortem and corresponded to the granular layer of the DG. All electrodes were attached to a pedestal (Bilaney Consultants GmbH) and secured with dental acrylate (Duracryl Plus, Spofa Dental, Czech Republic). Before vEEG monitoring, animals were separately placed in plexiglass cages and connected to a digital acquisition system with commutators (SL6C, Plastic One). vEEG was continuously carried out with TWin Clinical Software for EEG (Grass Technologies, Middleton, WI, USA) and I-PRO WV SC385 digital cameras (Panasonic, Kadoma, Japan) for 4 weeks (24/7).

### 2.6. Analysis of Video EEG Recordings

The occurrence of spontaneous seizures was analyzed by manually browsing through EEG recordings on a computer screen. An encephalographic episode was defined as a high amplitude (over 2× baseline) discharge lasting ≥10 s. The behavioral severity of each detected spontaneous seizure was analyzed from the matching video recordings. Seizure severity was estimated according to Racine [52], Schauwecker and Steward [53], and Ndode-Ekane and Pitkanen [54] with minor modifications: grade 0—encephalographic seizure with no detectable motor manifestation; grade 1—rigid posture or immobility; grade 2—stiffened, extended, and often arched tail; grade 3—partial body clonus, head bobbing, or whole-body clonic seizures; grade 4—rearing, severe whole-body continuous clonic seizures while retaining posture; grade 5—continuous rearing and falling, with severe clonic seizures; grade 6—generalized tonic-clonic seizures with jumping. Spontaneous seizures were defined as episodes appearing after 24 h post-KA administration. During the analysis, various parameters were characterized: seizure latency (time to the first spontaneous seizure), number of seizures per day, seizure score (behavioral severity), and average seizure duration.

### 2.7. Immunofluorescence

Mice were transcardially perfused with 4% *w*/*v* paraformaldehyde (PFA; Sigma-Aldrich) in phosphate-buffered saline (PBS), and brains were dissected and post-fixed by immersion in the same buffer overnight at 4 °C. Then, they were cryoprotected in 30% *w*/*v* sucrose in PBS for 24 h at 4 °C, snap-frozen in isopentane, and stored at −70 °C. After cutting 40-µm-thick coronal brain slices, they were subjected to antigen retrieval for 30 min at 80 °C in a preheated sodium citrate buffer (10 mM sodium citrate, 0.05% Tween 20, pH 6.0). Subsequently, slices were washed in PBS, blocked in 5% *v*/*v* NDST (normal donkey serum in PBS + 0.2% Triton X-100) for 1 h at room temperature (20–24 °C, RT), and incubated with primary antibody diluted in 5% NDST overnight at 4 °C. This step was followed by a PBS wash and incubation with species-specific secondary antibodies in 5% NDST for 2 h at RT. After incubation, slices were washed, co-stained with Hoechst (Invitrogen, Carlsbad, CA, USA, H3570; 1:500), a cell-permeant nuclear stain, for 10 min at RT, washed again, and coverslipped on glass slides with Vectashield (Vector Laboratories, Newark, CA, USA). Primary antibodies used: sheep anti-CD44 (R&D Systems, AF6577; 1:500), chicken anti-glial fibrillary acidic protein (GFAP) (Abcam, ab4674; 1:500), and rabbit anti-s100β (Abcam, ab52642; 1:100). Secondary antibodies used: donkey anti-sheep conjugated to Alexa Fluor^TM^488 (Invitrogen; 1:500), donkey anti-sheep conjugated to Alexa Fluor^TM^555 (Invitrogen; 1:500), donkey anti-rabbit conjugated to Alexa Fluor^TM^647 (Jackson ImmunoResearch; 1:500).

### 2.8. Fluorescence Microscopy Analysis

Microscopic imaging of fluorescent specimens was carried out using a Leica AF 7000 fluorescence microscope (objective lenses: HCX PL APO 10.0×/0.40 DRY, HCX PL APO 20.0×/0.70 DRY) and a Zeiss LSM 800 with Airyscan confocal microscope (objective lenses: EC Plan-Neofluar 10×/0.30, Plan-Apochromat 63×/1.40 Oil DIC M27). All scans were analyzed with Fiji software. Fractions of virally transfected astrocytes were calculated from s100β-immunostained slices. The area of astrocytes was determined by outlining individual astrocytes observed as GFAP immunosignal. Because GFAP protein is mostly localized in the major primary and secondary branches [55,56], undetectable terminal astrocyte processes were excluded from this analysis. Comparative analyses were only performed in the case of images taken with the same acquisition parameters.

### 2.9. SBEM Sample Preparation

Mice were transcardially perfused with 4% PFA + 2.5% e/v glutaraldehyde (GA; EM grade, Sigma-Aldrich) in PBS. Brains were dissected and post-fixed by immersion in the same buffer overnight at 4 °C, then cut on a vibratome into 100 µm-thick coronal slices. The ML of the DG region in the hippocampus was cut out from each slice, along with a small piece of granular layer for orientation, with a razor blade.

SBEM staining was performed based on a previously published protocol [57] with slight modifications. Specimens were washed in cold 0.1 M phosphate buffer pH 7.4, postfixed in 1.5% potassium ferrocyanide (Sigma-Aldrich) and 2% osmium tetroxide (Electron Microscopy Sciences, Hatfield, PA, USA) aqueous solution for 1 h on ice. All later steps were performed at room temperature. Samples were washed with double distilled (dd) H_2_O, immersed in 1% aqueous thiocarbohydrazide (TCH; Sigma) solution for 20 min, and washed again. Subsequently, they were postfixed in a 2% osmium tetroxide aqueous solution for 30 min, rinsed with dd H_2_O, and incubated in a 1% aqueous solution of uranyl acetate overnight. The next morning, samples were washed and immersed in a freshly prepared lead aspartate aqueous solution pH 5.5 (0.066 g lead nitrate in 10 mL dd H_2_O + 0.04 g aspartic acid pH 3.8) for 20 min at 60 °C, then washed again. Next, they were dehydrated using reversely graded dilutions of ethanol (20%, 50%, 70%, 100%, 100% anhydrous) for 5 min each. After the last ethanol incubation, fragments were infiltrated with a 1:1 (*v*/*v*) solution of Durcupan resin (Sigma-Aldrich) with anhydrous ethanol for 30 min, and 100% Durcupan resin for another 1 h. The resin was then replaced with a fresh one and left to incubate overnight. The next day, Durcupan resin was replaced again for an additional 1-h infiltration, after which samples were flat embedded between Aclar sheets (Electron Microscopy Sciences), and put in an oven at 65 °C for at least 48 h for the resin to polymerize. After the resin hardened, Aclar sheets were separated and the resin-embedded ML pieces were taken out, glued to aluminum pins (Gatan 3View system scanning electron microscopy pin stubs, Micro to Nano), and trimmed on an ultramicrotome (Leica Ultracut R). Samples were covered with conductive silver paint (Ted Pella) and mounted into the 3View chamber (Gatan).

### 2.10. SBEM Imaging

Imaging of samples was conducted with a SigmaVP (Zeiss, Oberkochen, Germany) scanning electron microscope equipped with the 3View2 chamber (Gatan, Pleasanton, CA, USA) using a backscatter electron detector. Scans were taken in the inner molecular layer of the hippocampus. From every sample, 200 sections were collected (thickness 60 nm). Imaging settings: 2.5 kV EHT, 20 µm aperture, 7–9 Pa variable pressure, 15,000× magnification, 7 µs pixel dwell time, 5 nm × 5 nm pixel size (3072 × 3072 camera array).

### 2.11. SBEM Analysis

Scans were aligned using the TrakEM plugin in Fiji software and normalized with Microscopy Image Browser 2.82 software [58]. The reconstruction of synaptic connections and astrocytic leaflets was carried out using Reconstruct 1.1.0.0 software [59]. For 3D reconstruction, areas from each sample taken for analysis were randomly picked and processed using the unbiased brick method [60]. For every specimen, reconstructed structures were limited to a brick of size 6 µm × 6 µm × 6 µm. Dendritic spines were defined as dendritic protrusions having an electron-dense region, the approximate core of post-synaptic density (PSD), close to the membrane adjacent to an axonal bouton with synaptic vesicles. Astrocytes and their leaflets were identified by characteristic features: an irregular stellate shape, numerous glycogen granules, bundles of intermediate filaments, or a relatively clear cytoplasm. After identifying and reconstructing PSD areas, dendritic spines and axonal boutons, the parts of astrocytic leaflets located in their proximity, were drafted. Dendritic spines were classified as filopodia (protrusions without enlargement at the tip), stubby spines (short protrusions without a neck and in direct continuity with the dendritic surface), and mushroom spines (protrusions with a neck and an enlargement at the tip). The PSD surface area and volume were measured by outlining darkened electron-dense regions on every section with a particular PSD. Following the work of Ostroff et al. [61], an astrocytic leaflet localized in the proximity ≤20 nm from a PSD was considered as one having contact with the synapse. Reconstruct software was used to extract PSD parameters and PSD to leaflet distance.

### 2.12. Statistical Analysis and Figures

Data statistical analysis was performed using GraphPad Prism 9 software (GraphPad Software Inc., San Diego, CA, USA). Distributions were tested with the Shapiro-Wilk normality test. Differences in semiquantitative scores between groups were analyzed using Fisher’s test. For other comparisons, the parametric Student’s *t*-test (groups with equal variances), Student’s *t*-test with Welch’s correction (groups without equal variances), two-way ANOVA, or the nonparametric Mann–Whitney test (groups with equal variances), the Kolmogorov-Smirnov test (groups without equal variances) were used. Data on graphs are presented as mean ± standard errors of the mean (SEM) in parametric analyses or as median with interquartile range (IQR) in nonparametric analyses. Differences between groups were considered significant if *p* < 0.05 (* *p* < 0.05, ** *p* < 0.01, *** *p* < 0.001). The Pearson correlation coefficient was calculated to measure the correlation between the intensity of CD44 and the area of GFAP immunoreactivities.

Apart from the information on the statistical parameters included in the figure captions, additional parameters (Appendix A) were calculated in Python (ver. 3.7.4) using the following libraries: numpy (1.19.2) and scipy (1.5.2). The linear regression was performed using the linregress module (1.10.1). The power analysis was performed using a corresponding function from the statsmodels module (0.12.1). It was calculated for a pair of datasets based on the number of data points of a larger set. The effect size was calculated as a difference of means divided by the minimum value of standard deviations. The significance level was 0.05.

## 3. Results

### 3.1. Increased Level of CD44 in Astrocytes following Kainate-Induced Status Epilepticus

To shed light on the role of CD44 in the development and progression of epilepsy, we characterized the expression of CD44 in the mouse hippocampus after kainic acid-induced seizures. *Status epilepticus* was induced in CD44*^fl/fl^* mice by intrahippocampal KA injections (*n* = 4 per group). This model reproduces many of the morphological and functional features observed in human patients with temporal lobe epilepsy [51]. The immunohistochemistry with anti-CD44 and anti-glial fibrillary acidic (GFAP) antibodies was performed to visualize CD44 protein expression in astrocytes (Figure 1a,b). We compared the CD44 level in the hippocampus of control animals and animals 4 days after KA-induced SE. Sham injections with saline (SA) were used to give sufficient control for KA administration in terms of the mechanical injury caused by needle insertion and fluid infusion. The astrocytic GFAP profile area increased 4 days after KA-induced SE (Figure 1c). Furthermore, in comparison to a low level of CD44 expression in SA-treated controls, we observed a visible upregulation of CD44 in hippocampal astrocytes in the ML of the DG (Figure 1d).

### 3.2. CD44 Depletion in Astrocytes by AAV5-Mediated Expression of gfaABC1D::GFP-Cre Recombinase in the Hippocampi of CD44^fl/fl^ Mice

To further investigate the role of CD44 in astrocytes during epileptogenesis, we implemented a genetic approach to delete CD44 exclusively in the hippocampal astrocytes. We performed stereotaxic delivery of an adeno-associated viral vector (AAV5) encoding Cre-recombinase into the molecular layer of the dentate gyrus of the hippocampi of CD44^fl/fl^ mice (Figure 2a). Cre recombinase was expressed in astrocytic cell nuclei and was N-terminally appended with green fluorescent protein (GFP), a fluorescent reporter, under the control of an astrocyte-specific GFAP promoter. These mice are indicated in the figures as astrocytic CD44 knockouts (AsKO). As a control (indicated as CTRL), we used CD44fl/fl mice injected with the AAV5 virus encoding GFP under the control of the GFAP promoter, resulting in GFP expression in the cell cytoplasm of astrocytes. To estimate the efficiency of viral transduction, we calculated the percentage of s100β-positive cells that were also GFP-positive. The analysis revealed that the efficiency of transduction was 84.1 ± 3.51% for AAV5.gfaABC1D::GFP and 74.22 ± 3.07% for AAV5.gfaABC1D::GFP-Cre (data presented as mean ± SEM). Next, we examined CD44 expression levels by Western blot analysis, 4–5 weeks after AAV delivery. We found a strong reduction of CD44 in the DG of mice expressing Cre recombinase compared to the controls (Figure 2b,c). Immunohistochemical analysis revealed that, unlike the CTRL group, AsKO animals had decreased CD44 expression in the infected astrocytes (detected with the anti-s100β antibody) in the molecular layer of the DG (Figure 2d,e).

### 3.3. CD44 Deficiency Reduces KA-Induced Epileptogenesis

To investigate the role of CD44 in the process of epileptogenesis, mice with CD44 deficiency in hippocampal astrocytes and CTRL animals (*n* = 12 per group) were intrahippocampally injected with KA, and continuously vEEG monitored for the next 4 weeks (Figure 3a). Both groups developed seizures during the recording period. When analyzing seizures with behavioral manifestations (grades 1–6; for scale description see Section 2.6 of Materials and Methods), no significant differences in the latency to the first spontaneous seizure (Figure 3b), the average duration of spontaneous seizures (Figure 3c), or seizure severity (Figure 3d) were observed. However, the average number of behavioral seizures per day during the fourth week was significantly lower in CD44 AsKO relative to the CTRL animals (Figure 3e). In contrast, the average number of grade 0 seizures (detected in EEG but without any behavioral symptoms) was significantly higher in knockout mice than in the controls (Figure 3f); 84% of all seizures in the CD44 knockout group were of grade 0, compared to only 33% in the control mice (Figure 3g). The difference in the total number of all seizures (including non-behavioral) between AsKO+KA and CTRL+KA animals (Figure 3g) mainly results from a significantly greater number of seizures of grade 0 observed in AsKO+KA mice (Figure 3f).

### 3.4. CD44 Deficiency Reduces Reactive Astrogliosis after Kainate-Induced Status Epilepticus

It is well known that reactive astrogliosis occurs in the mouse hippocampus in response to seizures (rev. in [62]). Having demonstrated that CD44 is upregulated in astrocytes following SE, we then evaluated how CD44-deficient astrocytes respond to kainate-induced SE. To assess reactive astrogliosis, we performed GFAP immunohistochemical staining in CTRL and AsKO mice 4 weeks after SA (Figure 4a) or KA injection (Figure 4b). For quantification, the mean area occupied by GFAP immunoreactivity (GFAP-ir) for a single astrocyte, which reflects the extent of reactive astrogliosis, was measured. The results showed that CD44 deletion had no significant effect on the GFAP-ir area in naive, saline-treated mice (Figure 4c), whereas seizure-induced astrogliosis was significantly reduced in CD44 AsKO+KA versus CTRL+KA mice (Figure 4e). Alongside the GFAP immunostaining, we additionally assessed the CD44 expression level in astrocytes. CD44 immunoreactivity (CD44-ir) was lowered in AsKO vs. CTRL mice (Figure 4d). The seizure-induced upregulation of CD44 at this time point was still well observed in the CTRL+KA group (Figure 4b,f) and was even more robust than the one seen 4 days post-kainate administration (presented in Figure 1). There was no apparent CD44-ir upregulation in AsKO+KA compared to untreated AsKO (Figure 4d,f). Moreover, the intensity of CD44-ir positively correlated with the area of GFAP-ir in both the CTRL (R = 0.901, Pearson correlation coefficient) and in CTRL+KA (R = 0.919, Pearson correlation coefficient) groups. The correlation was not observed in the AsKO group (R = 4.8 × 10^−5^, Pearson correlation coefficient) or was very low in the AsKO+KA group (R = 0.52, Pearson correlation coefficient) (see Appendix A for more details of the analysis), probably due to the low CD44 expression in the AsKO.

### 3.5. Ultrastructural Changes in Astrocyte-Synapse Interactions and Dendritic Spines in CD44 AsKO Mice Hippocampi upon Seizures

To establish if alterations in seizure severity in CD44 astrocytic knockouts in the hippocampus are accompanied by structural changes in the astrocyte-neuron relationship, we employed electron microscopy and characterized synapses and adjacent astrocytes at the ultrastructural level. We used SBEM to three-dimensionally reconstruct synaptic contacts together with surrounding astrocytic terminal processes from the ML of the DG. All structures were analyzed from tissue bricks of 6 µm × 6 µm × 6 µm size per animal following the unbiased brick method [60]. We performed a volumetric analysis of PSDs representing the postsynaptic part of the synapse. We reconstructed 192 synapses with adjacent astrocyte leaflets from the control group (CTRL+KA) and 472 from the AsKO mice (AsKO+KA) upon KA-induced seizures. The distances between reconstructed PSDs and adjacent astrocytic leaflets in the ML of the hippocampi of CTRL+KA and AsKO+KA mice were measured (Figure 5a,b). Adopting the methodology of Ostroff et al. [61], we assumed that a leaflet-PSD distance ≤20 nm represents synapses with astrocyte contacts. As a result, we observed a lower percentage of such contacts in AsKO+KA mice in comparison to the CTRL+KA group (Figure 5c). Since extensive spine loss in the dendrites of the DG granule cells has been previously shown upon kainate-induced seizures [16], we also analyzed changes in dendritic spine density in mice after SA (CTRL and AsKO) or KA (CTRL+KA and AsKO+KA) injection. The density of dendritic spines in the ML was reduced upon seizures in CTRL, but not in AsKO mice (Figure 5d–f). At the same time, the PSD mean volume and area were significantly diminished in AsKO+KA mice in comparison to CTRL+KA animals (Figure 5h,i). To more precisely define the characteristics of the dendritic spines, we classified them into three main categories: filopodia, stubby spines, and mushroom spines. The ratio between these different types of dendritic spines did not vary in the CD44 AsKO+KA vs. CTRL+KA groups (Figure 5g). Together, these data show that the depletion of CD44 in astrocytes leads to a reduced interaction of astrocytic leaflets with the synaptic cleft, an increased density of dendritic spines, and a decreased size of postsynaptic densities in the epileptic ML within the dentate gyrus of the hippocampus.

## 4. Discussion

In the present study, we report that mice with selective deletion of CD44 in hippocampal astrocytes developed less severe epileptogenesis and gliosis upon intrahippocampal KA administration. Furthermore, we characterized ultrastructural seizure-induced changes in dendritic spines and thin astrocytic processes that contact synapses in the DG of the hippocampus of the CTRL and CD44 AsKO animals. Our work revealed that, compared to the control mice, astrocytic deficiency of CD44 in the epileptic hippocampus leads to a diminished astrocytic coverage of synapses, an increased dendritic spine number, and a decreased size of postsynaptic densities.

CD44 has been previously shown to be predominantly expressed in astrocytes in rodents and the human brain, and it has been demonstrated to be important in the regulation of astrocyte morphology [39,58]. CD44 expression increases in the brain after injury [29,63]. However, there are inconsistent data concerning its expression in animal models of epilepsy. CD44 was strongly upregulated in the dentate gyrus molecular layer 3 days post-pilocarpine-induced SE in mice, then it declined over the next 4 weeks [42]. This adhesion molecule was also identified in a proteomic approach as one of the most robustly induced proteins in the mouse brain in the kainate model of epileptogenesis [64]. Contrary to the above observation, studies in in vitro hippocampal organotypic cultures have shown that CD44 expression in the molecular layer was reduced following kainic acid treatment [40]. In our model of TLE, CD44 was overexpressed in hippocampal astrocytes and this enhanced expression persisted for the next four weeks. A potentially critical difference between studies may arise from the use of different models of epilepsy. The cited study was conducted in cell culture under strictly defined conditions [40], whereas ours was carried out in the in vivo mouse model where a variety of systemic players may influence the observed alterations. Although it is important to take into consideration both in vitro and in vivo results, the latter in many cases better represents the complicated changes happening in a living organism. Increased CD44 immunoreactivity in the astrocytic processes was also shown in human patients with tuberous sclerosis [65] and the mouse model of Alexander disease [58], which are conditions accompanied by seizures. Recently, CD44 was identified in genome-wide transcriptome analysis as one of the treatment targets for epilepsy in rats [44] and humans [45].

A growing body of evidence indicates that the elevated expression of CD44 is associated with epilepsy-related brain injuries [42,44,45,58,64,66]. However, the exact role of CD44 in activated astrocytes in epileptogenesis remains to be determined. Astrocytes are increasingly recognized as playing a fundamental role in bidirectional communication with neurons. They modulate neuronal excitability, synaptic transmission and plasticity [67], although the molecular mechanisms underlying these processes are largely unknown. Thus, we employed a viral infection approach to specifically target CD44 in hippocampal astrocytes to unravel its function in epileptogenesis. Based on the existing literature data [68,69] and our preliminary tests, we chose the AAV5 viral vector serotype and the astrocyte-specific GFAP promoter to obtain cell-specific CD44 deletion. Our results indicate that injection of the AAV5 vector expressing Cre recombinase appended at its N-terminus by GFP under the control of GFAP promoter into the molecular layer of the DG of CD44*^fl/fl^* mice effectively reduces the expression of CD44 in hippocampal astrocytes. However, we observed that upon kainate treatment, GFP was also expressed in granule neurons of the DG. These results suggest that AAV5 can also infect DG neurons and, what is more important and surprising, is that kainic acid influences neuronal gene expression so strongly that it can induce expression controlled by a GFAP promoter in granule neurons. Our observation is in line with previous data showing the expression of GFAP-driving transgenes in hippocampal neuronal cells [70]. An alternative explanation of KA-induced GFAP promoter-driven expression in neurons could be the enhanced neurogenesis in the DG and survival of the new granule neurons. It has been previously shown that seizure activity leads to transiently increased cell proliferation in the DG during the first 2 weeks after *status epilepticus* [71]. New neurons in the DG arise from a population of radial neural stem cells that, like astrocytes, express GFAP [72,73] and survive for at least 6 months [74]. Therefore, we cannot completely exclude a neuronal contribution to the results observed in our study. However, taking into account that the level of CD44 in neurons of the adult brain is low and that KA induces CD44 expression in astrocytes but not in neurons, we can still assume that the observed effects are predominantly caused by the CD44 knockout in astrocytes rather than neurons.

Recently, the non-specific actions of Cre recombinase in the brain have been reported, but they regard mainly neurons, not astrocytes. The mechanism of the possible noxious effects of AAV-mediated Cre expression described in these reports points to defects in DNA repair machinery that lead to a massive decrease in the neuronal population [75]. Adding WT mice with a Cre injection as an additional control could shed more light on this phenomenon. Since we focused our project on astrocytic CD44 and we did not observe massive cell death, we decided not to include the WT control injected with AAV5.*gfaABC1D*::GFP-Cre in our experiments to minimize the number of sacrificed animals. It has been previously shown that AAV viruses expressing eGFP controlled by the *gfaABC1D* promoter induce a titer-dependent reactivity of astrocytes [76]. Following low viral-titer injection (10^10^ genome copies (GC) per injection), eGFP-positive cells had normal astrocytic morphology and low levels of GFAP staining, whereas eGFP-positive cells labeled by high-titer injection (3 × 10^10^ genome copies per injection) of the virus had a hypertrophic morphology typical of reactive astrocytes. In our experiments, we used a low viral titer injection (CTRL: 4 × 10^9^ and Cre: 2.2 × 10^9^ genome copies (GC) per injection) that does not induce astrocytic gliosis.

In the present study, CD44 AsKO mice developed fewer spontaneous behavioral seizures following KA-induced *status epilepticus* than CTRL mice. The majority of the seizures observed in CD44 knockouts were of grade 0; i.e., were detected with EEG recording but expressed no behavioral symptoms. Thus, our findings demonstrate that CD44 depletion in astrocytes leads to defects in behavioral seizure development, but the molecular mechanisms of CD44 action are still unknown. Epilepsy is linked to a robust aberrant synaptic plasticity occurring in the ML of the DG [77]. Recently, it was determined that astrocytic deletion of one of the CD44 binding partners, the actin-associated protein Ezrin, resulted in smaller astrocyte territories and reduced astrocytic coverage of excitatory synapses. This led to a decrease in the synaptic glutamate level and increased extrasynaptic glutamate diffusion, and had a significant impact on contextual fear memory [78]. Since the C-terminal cytoplasmic domain of CD44 interacts with Ezrin (reviewed in [79,80]), the observed changes in seizure severity in CD44 knockout animals might be explained by alterations in glutamate uptake and diffusion. This hypothesis is also supported by the results showing that increased CD44 in astrocytes correlates with a loss of plasmalemmal glutamate transporter GLT-1 and diminished glutamate transporter current [58]. It is possible that astrocytes with increased expression of CD44 upon KA-induced *status epilepticus* do not buffer glutamate as efficiently as non-reactive astrocytes.

Reactive astrocytes play a crucial role in CNS pathophysiology. Astrogliosis, the process of astrocyte activation upon brain injury (including epilepsy), is connected with changes in astrocytic morphology. We have previously shown that CD44 regulates astrocyte shape in primary astrocytic cultures [39]. In the present study, we observed reduced GFAP-ir profiles of astrocytes upon seizures in CD44 AsKO mice, indicating that the enhanced CD44 expression supports the development of astrogliosis. These results are in line with the previously published observation showing that phenotypic conversion from regular protoplasmic to hypertrophic astrocytes in Alexander disease is associated with CD44 acquisition by normally CD44-negative protoplasmic astrocytes. This suggests that CD44 can play a role in astrogliosis; e.g., by mediating alterations in astrocyte shape [58]. There is also evidence that astrogliosis can be a cause of seizures. Genetically- or virally-induced astrogliosis was shown to be sufficient to cause epileptic seizures in mice without any other CNS pathologies [76,81]. Thus, CD44 might also regulate seizure severity by contributing to the development of astrogliosis.

Epilepsy results from an imbalance between neuronal excitation and inhibition. Among effective mechanisms for decreasing epilepsy occurrence are (i) the conversion of polyamines to GABA and (ii) the subsequent release of GABA from astrocytes [82,83,84]. There is no direct evidence that CD44 function is related to GABA release. Nevertheless, the observed beneficial effect of CD44 deficiency on seizure progression may be associated with changes in the interplay between either CD44 itself or CD44-dependent signaling pathways and astrocytic GABA release or mechanisms regulating polyamine conversion to GABA in astrocytes. In addition to excitatory synapses, the astrocytic deletion of CD44 may as well alter inhibitory synapses made by GABAergic neurons that are also present in the molecular layer of the hippocampus.

Astrocytic leaflets are linked to synaptic ensheathment, and interaction with dendritic spines and synaptic terminals, which regulate synaptic structure and functions [21,85,86]. The interplay between neurons and astrocytes is one of the mechanisms that contribute to the generation of epileptic discharges [87]. Our results revealed that astrocytic deletion of CD44 during epileptogenesis has an impact on the structural relationship between astrocyte processes and synapses, leading to a decreased number of synapses with astrocyte contacts. This may be explained by the possibility that AsKO results in decreased astrogliosis. In line with this, the loss of Ezrin reduces astrocyte-neuron proximity and the number of astrocyte contacts at synapses, and these structural changes are accompanied by increased glutamate spillover and NMDA receptor activation [78]. Thus, it is likely that CD44-dependent changes in astrocytic synaptic coverage in CD44 AsKO mice might be associated with a reduced glutamate concentration at synapses and in turn lead to reduced seizure progression. On the other hand, in CTRL+KA animals, the glutamate concentration at the synapse can be increased by (i) the reduced level of GLT-1 in reactive astrocytes (reviewed in [88]), leading to an inhibition of glutamate reuptake by astrocytes at the synaptic cleft, and (ii) an increased number of astrocyte-synapse contacts, resulting in a decrease in glutamate spillover outside the synaptic cleft.

We showed that spine density significantly decreases in the ML of the dentate gyrus of CTRL mice upon KA administration. These results are in line with previously shown extensive spine loss in the dendrites of granule cells upon kainate-induced seizures [16]. In CD44 AsKO, we did not observe KA-induced spine loss. This may suggest a protective effect of the astrocytic CD44 deletion on the KA-induced loss of existing synapses in the hippocampus of knockout mice. Moreover, electron microscopy ultrastructural observations of the epileptic brain uncovered the presence of degenerated presynaptic profiles partially or fully enveloped in astrocytic leaflets [18,27,89], suggesting that seizures can induce active synaptic phagocytosis by reactive astrocytes. Therefore, the observed elevated number of synapses in the hippocampus of CD44 knockouts might be associated with reduced phagocytosis by CD44-depleted astrocytes. Increased dendritic spine density and decreased PSD size in CD44 AsKO animals upon seizures indicate that CD44 deletion from astrocytes can have beneficial effects on brain structure after epilepsy-related injury. One possible target is the prevention of the formation of aberrant giant spines associated with epileptic trauma [16,60], which are believed to be a compensatory mechanism for a reduced number of axonal boutons as a way to recover some of the lost excitatory drive [90]. This possibility is unlikely in our animal model because we did not observe such giant spines with sufficient frequency to be statistically significant (Figure 5g). Another possible target is the transition of newly formed spines, which are usually thin and often called immature or learning spines, to become mature mushroom spines, also known as memory spines [91,92,93,94]. Such morphological changes in dendritic spines are prerequisites of synaptic plasticity [95,96]. However, our study revealed that the proportion of dendritic spines of different shapes is not changed in the ML of the DG in the CD44 AsKO brain upon KA-induced seizures, indicating that all investigated types of dendritic spines are equally susceptible to the beneficial effect of astrocytic CD44 deletion.

## Figures and Tables

**Figure 1 cells-12-01483-f001:**
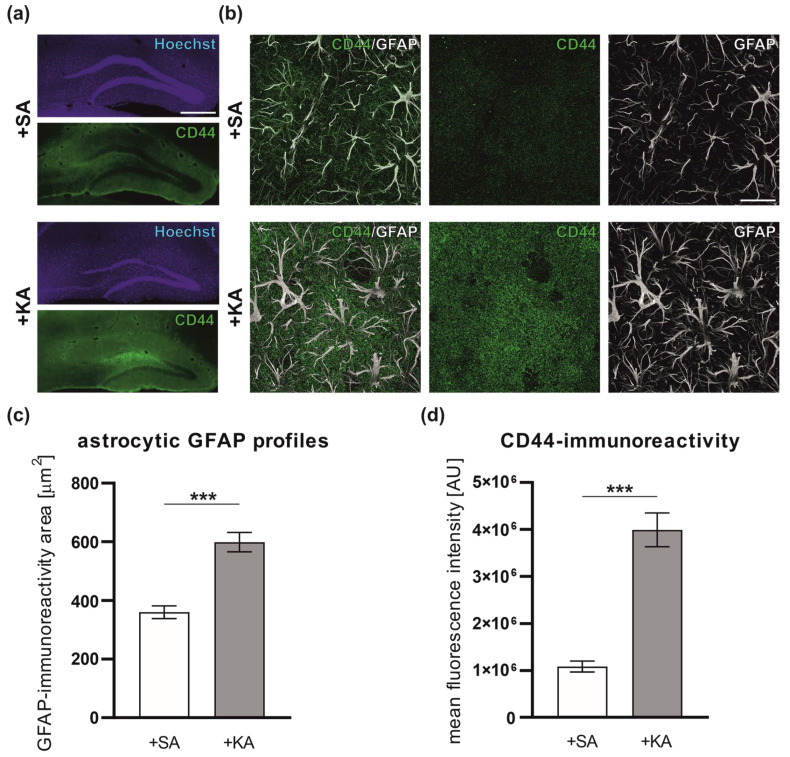
Upregulation of the CD44 expression in hippocampal astrocytes of CD44*^fl/fl^* mice following kainic acid (KA)-induced *status epilepticus*. (**a**) Representative immunofluorescence micrographs of CD44 expression (green) and nuclei (Hoechst, purple) in the hippocampus of SA- or KA-injected mice. Scale bar: 500 μm. (**b**) High-power view of the DG neuropil (molecular layer) immunostained for GFAP (white) and CD44 (green) in SA- or KA-injected mice. Scale bar: 20 μm. (**c**) Quantitative analysis of astrocytic GFAP area in CD44*^fl/fl^* mice 96 h after SA or KA administration (*n* = 4 independent animals in each group, Student’s *t*-test, *** *p* < 0.001, data presented as mean ± SEM) (**d**) CD44-immunoreactivity in astrocytic GFAP profiles 96 h after SA or KA administration (*n* = 4 independent animals in each group, Student’s *t*-test, *** *p* < 0.001, data presented as mean ± SEM).

**Figure 2 cells-12-01483-f002:**
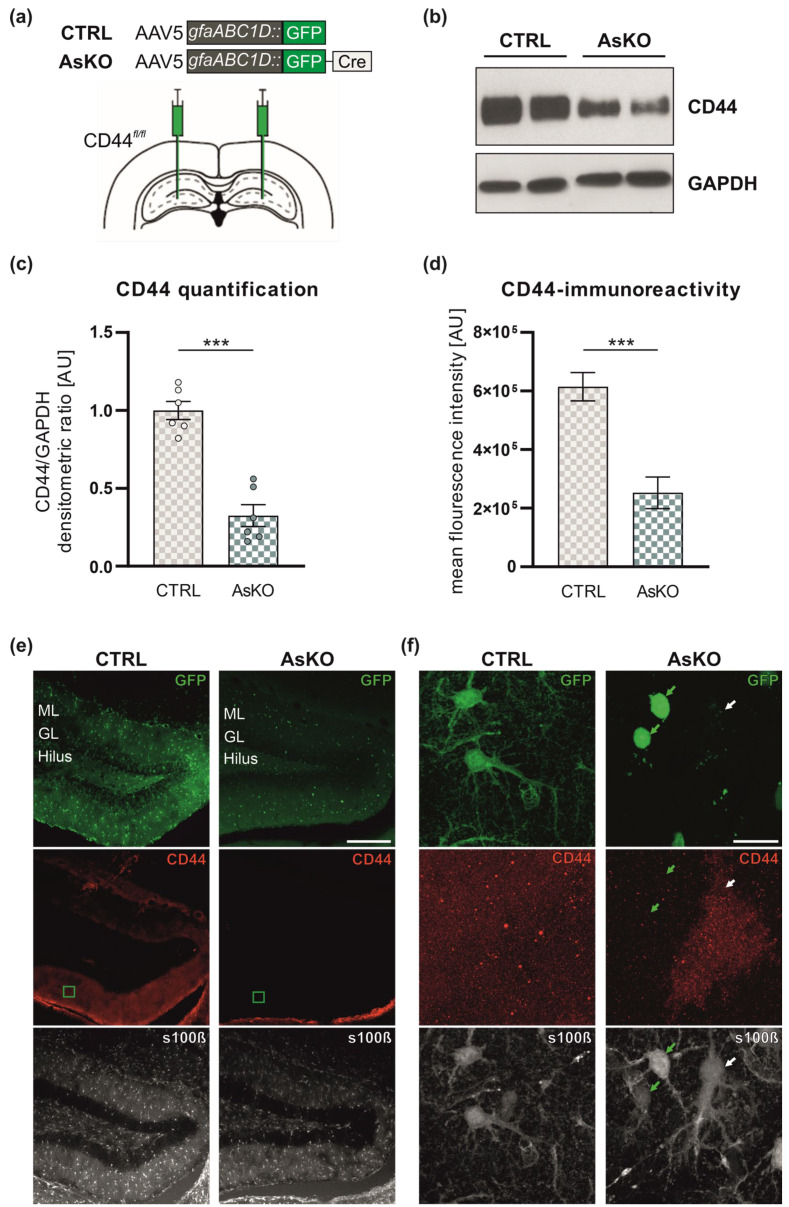
GFAP-Cre recombinase viral approach to target CD44 in adult dentate gyrus astrocytes. (**a**) Scheme of bilateral intrahippocampal administration of AAV vectors (CTRL: AAV5.*gfaABC1D*::GFP; GFP localizes in the cytoplasm of astrocytes, and AsKO: AAV5.*gfaABC1D*::GFP-Cre; GFP-Cre localizes in cell nuclei of astrocytes) into the brains of CD44^fl/fl^ mice. (**b**) CD44 level in protein extracts from DG of CTRL and AsKO animals (Western blot). Glyceraldehyde 3-phosphate dehydrogenase (GAPDH) was used as the loading control protein. (**c**) Quantitative analysis of Western blot results (*n* = 6 independent animals, Student’s *t*-test, *** *p* < 0.001, data presented as mean ± SEM) (**d**) CD44-immunoreactivity in astrocytic s100β profiles from the ML of the DG in CTRL and AsKO mice (see panel f for representative immunostaining) (Student’s *t*-test, *** *p* < 0.001, data presented as mean ± SEM). Representative immunofluorescence micrographs of GFP (green), astrocyte marker s100β (white), and CD44 (red) expression in the hippocampus of CTRL and AsKO mice for matching triple stain images shown at (**e**) a lower magnification or (**f**) the close-ups reflecting green squares on panel (**e**). Arrows point to astrocytes (s100β staining) that were or were not transfected (green arrow for GFP-expressing and white arrow for non-expressing, respectively). ML—molecular layer, GL—granular layer. Scale bar: (**e**) 200 μm and (**f**) 20 μm.

**Figure 3 cells-12-01483-f003:**
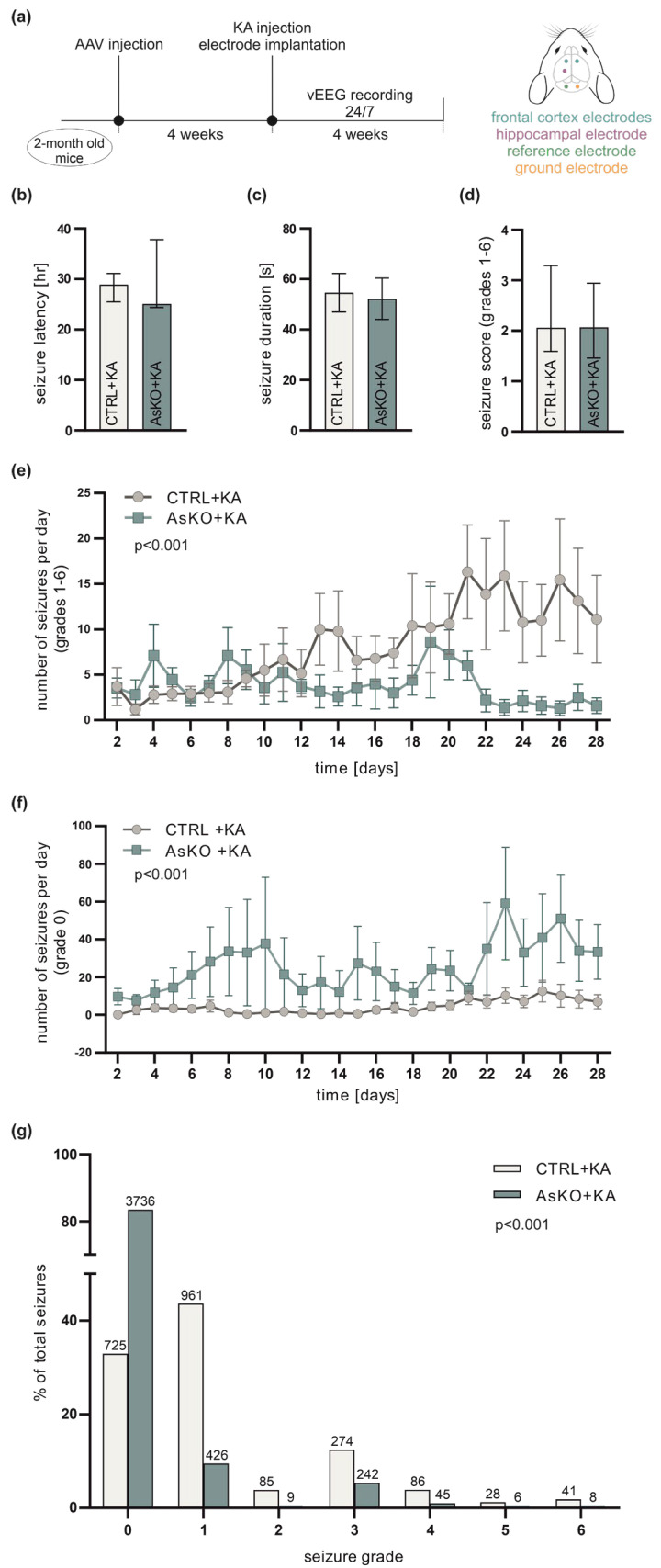
Astrocytic knockout of CD44 in the dentate gyrus results in fewer spontaneous behavioral seizures after intrahippocampal KA administration. (**a**) Schematic representation of the experimental design. (**b**) Seizure latency to the first spontaneous seizure was not affected by CD44 knockout (Mann-Whitney test, *p* = 0.22, data presented as median with IQR). (**c**) The mean duration of spontaneous seizures was not different in the AsKO compared to the CTRL group (Student’s *t*-test, *p* = 0.83, data presented as mean ± SEM). (**d**) The average seizure score (for seizures of grades 1–6) was not different in the AsKO compared to the CTRL group (Mann-Whitney test, *p* = 0.38, data presented as median with IQR). (**e**) Knockout animals developed a decreased number of behavioral seizures (grades 1–6) per day when compared to CTRL animals (two-way ANOVA, genotype effect *p* < 0.001, data presented as mean ± SEM). (**f**) CD 44 knockout animals developed a greater average number of seizures of grade 0 per day than CTRL mice (two-way ANOVA, genotype effect *p* < 0.001, data presented as mean ± SEM). (**g**) The severity of behavioral seizures presented as a percentage of each seizure type (grades 0–6) out of all episodes per group. Most seizures in AsKO mice were less severe than the ones occurring in the CTRL group (Fisher’s exact test, *p* < 0.001).

**Figure 4 cells-12-01483-f004:**
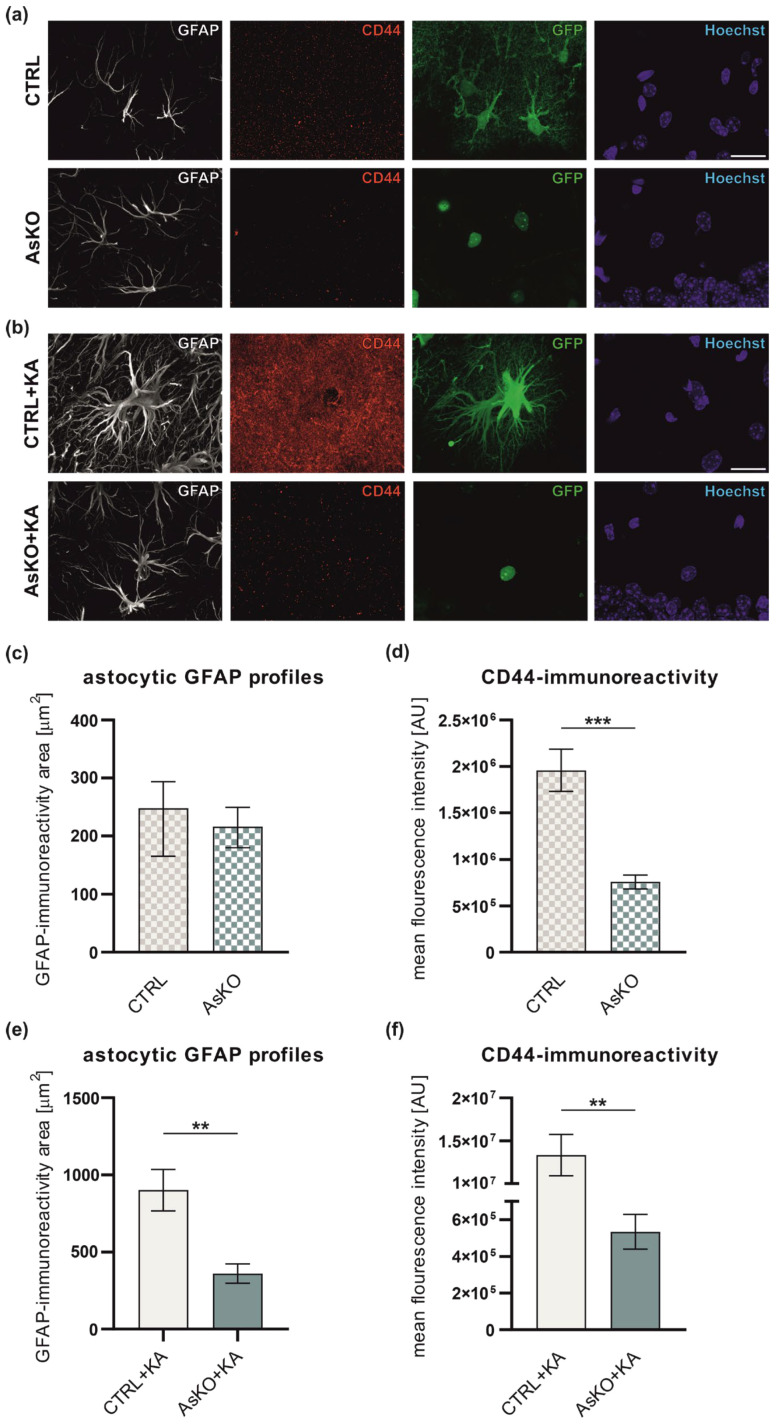
CD44 deletion in astrocytes reduces reactive astrogliosis following kainate-induced epilepsy. Representative images of astrocytes (GFAP, white), CD44 expression (red), GFP (green), and nuclei (Hoechst, purple) in the hippocampal ML of the DG of CTRL and CD44 AsKO mice before (**a**) and after (**b**) 4 weeks post-KA administration. Scale bar: 20 μm. Quantitative analysis of GFAP-immunoreactive area (µm^2^) in the ML of the DG in CTRL and CD44 AsKO mice 4 weeks after administration of (**c**) SA (Kolmogorov-Smirnov test, *p* = 0.41, data presented as median with IQR) or (**e**) KA (*n* = 6 independent animals in each group, Student’s *t*-test with Welch’s correction, ** *p* = 0.004, data presented as mean ± SEM). Quantitative analysis of CD44-immunoreactivity in astrocytic GFAP profiles in the ML of the DG in CTRL and CD44 AsKO mice 4 weeks after administration of (**d**) SA (Student’s *t*-test with Welch’s correction, *** *p* < 0.001, data presented as mean ± SEM) or (**f**) KA (Student’s *t*-test with Welch’s correction, ** *p* = 0.001, data presented as mean ± SEM).

**Figure 5 cells-12-01483-f005:**
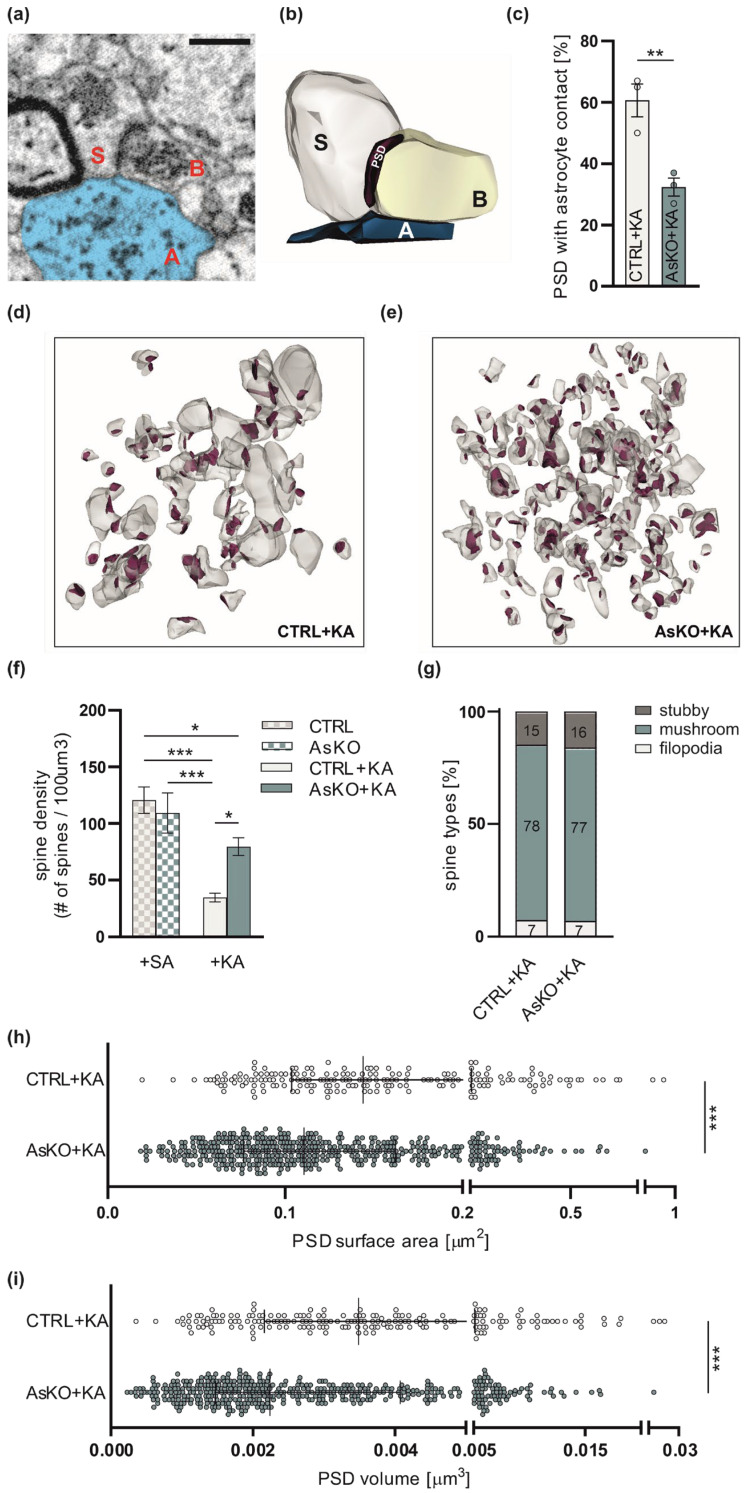
Astrocytic knockout of CD44 in the ML of the dentate gyrus prevents KA-induced spine loss and results in reduced PSD volume and surface area and diminished astrocyte–synapse interaction in the epileptic brain. (**a**) Representative image of a synaptic structure from SBEM. Astrocytic leaflet (A, blue) contacting presynaptic bouton (B) and postsynaptic dendritic spine (S). Scale bar: 500 nm. (**b**) Example of a 3D reconstruction of the tripartite synapse structure with PSD. (**c**) The proportion of synaptic clefts with astrocyte contacts. Shown as a percentage of PSDs with astrocytic leaflets in a ≤20 nm proximity (Fisher’s exact test, ** *p* < 0.001, data presented as mean ± SEM). (**d**,**e**) Examples of 3D reconstructions of spines (white) with PSDs (red) from CTRL+KA and AsKO+KA mice (brick size: 6 µm × 6 µm × 6 µm). (**f**) Spine density was decreased in CTRL mice following KA treatment. In CD44 AsKO animals the KA-induced reduction in spine density is diminished (two-way ANOVA, * *p* < 0.05, *** *p* < 0.001 data presented as mean ± SEM). (**g**) The proportion of protrusion types (mushroom spines, stubby spines, and filopodia) detected on dendritic segments imaged with SBEM (two-way ANOVA). (**h**,**i**) Both PSD area and volume were decreased in AsKO+KA mice (Mann-Whitney test, *** *p* < 0.001, data presented as median with IQR).

## Data Availability

Raw Data that support the findings of this study are available from the corresponding author upon request.

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
