# Peer review of "Astrocytic CD44 Deficiency Reduces the Severity of Kainate-Induced Epilepsy"

_cells, 2023, doi:10.3390/cells12111483_

Round 1

Reviewer 1 Report

To attract more readers and to increase the number of citations of this very important manuscript the authors need to cover professionally in the introduction (and/or discussion) the important paragraphs to emphasize the role of the astrocytes in epilepsy supported by proper citations.

Specifically, the drug-resistant type of epilepsy (which occurs in about a third of the patient population) highlights the role of neuronal unacknowledged partners, the astrocytes.  Astrocytes play a key role in both quenching and accelerating seizures depending on many varieties. For these reasons, the authors need to write important paragraphs about the different roles of the astrocytes in epilepsy supported by proper citations (see below).

CD44 is an astrocytic representative found primarily in the white matter of the brain (McKenzie et al., 1982). Mostly, CD44 was found in fibrous astrocytes and perivascular astrocytes but not in protoplasmic (Naruse et al., 2013; Sosunov et al., 2014) and not in most adult neurons (Vogel et al., 1992, J. Neurocytol.).  Intriguingly, soon after birth, CD44 is greatly downregulated in some parts of the brain in astrocytes and neurons (Vogel et al., 1992, J.Neurocytol.; Akiyama et al., 1993, Brain Res.). This highlights the diversity of CD44 localization and function and needs to be stated.

In earlier studies, CD44 was found to be dramatically overexpressed in Alzheimer’s patient’s astrocytes (Akiyama et al., 1993, Brain Res.) however it is not related to epilepsy. Therefore, there is a mystery of CD44 expression, down-regulation, and normal function. In fact, CD44 is involved in a large number of functional links to many CNS disorders and could be a promising target in the treatment of epilepsy. Therefore, the authors need to emphasize such a challenge in the introduction.

Unfortunately, epilepsy can cause the loss of neurons in humans and animal models leaving fewer targets for astrocytes. Indeed, the extensive loss of small synapses with forming multisynaptic giant spines may limit astrocytic coverage and synaptic control because the surface area that astrocytes can cover is dramatically decreasing. In the abstract authors stated that there is a "reduced percentage of astrocyte-synapse contacts". Such contradiction needs to be explained somehow.

Among effective mechanisms for decreasing epilepsy occurrence are (i) the conversion of polyamines to GABA and (ii) the following release of GABA from astrocytes measured by mass spectrometry (Kovacz et al., 2022 Front. Cell Neurosci.). GABA released from astrocytes via GAT-2, GAT-3 transporters (iii) depresses synaptic excitatory activity in neurons by the mechanism of tonic inhibition (Kovacs et al., 2022, Biomolecules, MDPI). It is still unknown if CD44 function is related to GABA release. Authors need to highlight such mechanisms and a possible role of CD44 co-interaction with GABA release or with GAT2,3 transporters, or polyamine conversion to GABA in astrocytes.

The articles that need to be included in the reference list are the following:

Akiyama H, Tooyama I, Kawamata T, Ikeda K, McGeer PL. (1993) Morphological diversities of CD44 positive astrocytes in the cerebral cortex of normal subjects and patients with Alzheimer's disease. Brain Res. 632(1-2):249-59. doi: 10.1016/0006-8993(93)91160-t.

Héja, L.; Barabás, P.; Nyitrai, G.; Kékesi, K.A.; Lasztóczi, B.; Toke, O.; Tárkányi, G.; Madsen, K.; Schousboe, A.; Dobolyi, A.; et al. (2009) Glutamate uptake triggers transporter-mediated GABA release from astrocytes. PLoS ONE, 4, e7153.

Kovács, Z.; Skatchkov, S.N.; Veh, R.W.; Szabó, Z.; Németh, K.; Szabó, P.T.; Kardos, J.; Héja, L. (2022) Critical Role of Astrocytic Polyamine and GABA Metabolism in Epileptogenesis. Front. Cell. Neurosci. 15:787319. doi: 10.3389/fncel.2021.787319

Kovács, Z.; Skatchkov, S.N.; Szabó, Z.; Qahtan, S.; Méndez-González, M.P.; Malpica-Nieves, C.J.; Eaton, M.J.; Kardos, J.; Héja, L. Putrescine Intensifies Glu/GABA Exchange Mechanism and Promotes Early Termination of Seizures. Int. J. Mol. Sci. 2022, 23, 8191. https://doi.org/10.3390/ijms23158191

Reviewer 2 Report

The authors have studied the role of Astrocytic CD44 and its deficiency in an epilepsy induced animal model. The study has a very good approach, and the manuscript is well written. I would recommend publishing the manuscript after the following minor revisions.

- Please provide more details on the statistical analysis section.

- Data on graphs are recommended to be presented by Mean+SD instead of Mean+SEM, especially in the cases that n is small.
